# Impact of the COVID-19 Pandemic on Emergency Air Medical Transport of Pediatric Patients in the Penghu Islands

**DOI:** 10.3390/healthcare13121450

**Published:** 2025-06-17

**Authors:** Hung-Hsiang Fang, Chuang-Yen Huang, Po-Chang Hsu, Chia-Cheng Sung, Sheng-Ping Li, Chung-Yu Lai

**Affiliations:** 1Department of Pediatrics, Tri-Service General Hospital, National Defense Medical Center, No. 325, Section 2, Cheng-Kung Road, Neihu, Taipei 114, Taiwan; 2Department of Obstetrics and Gynecology, Tri-Service General Hospital, National Defense Medical Center, Taipei 114, Taiwan; 3Division of Pediatrics, Kaohsiung Armed Forces General Hospital, Kaohsiung 802, Taiwan; 4Division of Pediatrics, Tri-Service General Hospital Penghu Branch, National Defense Medical Center, Magong City 880, Taiwan; 5Graduate Institute of Aerospace and Undersea Medicine, National Defense Medical Center, Taipei 114, Taiwan; multi0912@gmail.com; 6Combat and Disaster Casualty Care Training Center, National Defense Medical Center, Taipei 114, Taiwan

**Keywords:** COVID-19 pandemic, emergency air medical transport, pediatric patients, air transportation, Penghu Islands

## Abstract

Background and Objectives: The coronavirus disease 2019 (COVID-19) pandemic significantly impacted healthcare systems worldwide. As a result, remote areas such as the Penghu Islands have encountered unique challenges related to pediatric care. This study examined the effects of the pandemic on the emergency air medical transport (EAMT) of pediatric patients from the Penghu Islands to Taiwan. Materials and Methods: This retrospective study analyzed 40 pediatric patients who received EAMT from the Penghu Islands to Taiwan between January 2017 and December 2022. This study compared patients before and during the COVID-19 pandemic and focused on patient demographics, reasons for EAMT, and clinical outcomes. Due to the small sample size, non-parametric statistical methods were applied, including the Mann–Whitney U-test for continuous variables and Fisher’s exact test for categorical variables. Results: Among the 40 pediatric patients analyzed, the median age decreased from 3 years (IQR, 0–5 years) before the pandemic to 1 year (IQR, 0–5 years) during the pandemic. While the overall increase in hospital length of stay during the pandemic was not statistically significant, a significant prolongation was observed in preschool-aged children and neonates without trauma (20 days vs. 9 days; *p* < 0.05). The lack of specialist physicians became an increasingly prominent factor for EAMT during the pandemic (*p* = 0.056). The most common medical reasons for EAMT were critical illness (35%), neonatal diseases (30%), and neurological conditions (27.5%), with similar distributions across both time periods. Conclusions: The COVID-19 pandemic heightened existing healthcare disparities in the Penghu Islands, particularly by increasing reliance on EAMT due to a shortage of pediatric specialists. Hospital stays for preschool children and neonates significantly increased during the pandemic, suggesting delayed or prolonged care. These findings underscore the need to strengthen local pediatric infrastructure, decentralize specialist services, and improve emergency preparedness to better support vulnerable populations in remote areas during future public health emergencies.

## 1. Introduction

In December 2019, an outbreak of pneumonia attributed to a novel coronavirus identified as severe acute respiratory syndrome coronavirus 2 (SARS-CoV-2) was reported in Wuhan, China [1,2]. Taiwan reported its first case of coronavirus disease 2019 (COVID-19) on 21 January 2020 [3]. The first in-hospital cluster infection occurred in late February 2020, thus marking the start of the peak epidemic period [4,5,6]. Between January 2020 and December 2022, Taiwan reported 8,872,955 confirmed COVID-19 cases [4]. In early 2020, Taiwan established the Central Epidemic Command Center to coordinate resources, develop effective policies, and implement strict interventions. In response to the COVID-19 pandemic, Taiwan enacted numerous infection control measures, including universal mask wearing, increased face mask production, hand hygiene measures, border control, digital technology integration with big data, quarantine procedures, and travel and social gathering restrictions [4].

Mainland Taiwan has numerous medical centers, regional hospitals, and district hospitals. In contrast, Penghu County has only three district hospitals that provide local medical services [7,8]. Penghu County is a large archipelago comprising 90 islands between China and Taiwan [7], and the Tri-Service General Hospital, Penghu Branch, is the primary healthcare facility for women and children. Most pediatric patients are referred to this hospital for further assessment and management.

Emergency transportation reflects the capacity and responsiveness of a healthcare system during a disease outbreak. During the COVID-19 pandemic, critical care transport services were required to rapidly adapt and develop operational approaches, protocols, and guidelines in quick succession [9]. Although previous studies have demonstrated that emergency air medical transport (EAMT) is an effective and safe intervention with various benefits for adults and children, the literature regarding the transfer of pediatric patients from Penghu is sparse [10,11,12,13,14,15,16].

The COVID-19 pandemic and its resulting protective measures have significantly affected hospital admissions for both respiratory and non-respiratory cases [17,18,19,20,21,22]. The previous medical literature indicated that during the COVID-19 pandemic, pediatric hospitalizations significantly decreased on an isolated island of Japan [23]. Specifically, there was a notable decrease in admissions attributable to respiratory infections during the height of the COVID-19 pandemic, followed by a rebound in cases after these measures were relaxed [17,22]. This study focused specifically on pediatric patients, who represent a vulnerable population requiring age-specific care, particularly in remote settings where pediatric subspecialists and critical care services may be limited. In regions like the Penghu Islands, emergency air medical transport is often essential for providing timely access to tertiary-level care for children with acute or complex conditions. Understanding the impact of the COVID-19 pandemic on this population is crucial for future healthcare planning and emergency response preparedness. Air transportation is a critical component of a region’s healthcare capacity and ability to respond to medical emergencies. However, limited research of EAMT for pediatric patients on remote islands during the COVID-19 pandemic has been conducted. This analysis focused on emergency cases of pediatric patients living in the Penghu Islands before and during the COVID-19 pandemic. Understanding the challenges and trends associated with pediatric EAMT during a public health crisis is essential for guiding future healthcare system planning. The findings from this study contribute to the broader understanding of healthcare system resilience and highlight the need for strategic emergency preparedness.

## 2. Materials and Methods

### 2.1. Study Design and Patients

This retrospective study was conducted by the Department of Pediatrics at Tri-Service General Hospital and Tri-Service General Hospital, Penghu Branch. During this study, we reviewed the electronic medical records and transport sheets of pediatric patients 18 years or younger who received EAMT from Tri-Service General Hospital, Penghu Branch, to Taiwan. This study was approved by the Institutional Review Board of Tri-Service General Hospital (IRB number: B202305029).

### 2.2. Data Collection

This study included 40 pediatric patients who were transferred to Taiwan via EAMT. Patients were eligible for inclusion if they received EAMT from Penghu County to Taiwan between January 2017 and December 2022. As this was a retrospective study, the sample size (n = 40) represents all eligible pediatric EAMT cases from Penghu to Taiwan during the study period, and no prior sample-size calculation was performed. Patients were excluded if complete documentation was missing, they discharged themselves from the hospital against medical advice, transportation was arranged by their family, or they travelled via a commercial flight to Taiwan. Clinical data including personal information, clinical details, season of the year when EAMT was performed, reasons for transfer, destinations, and diagnosed diseases were meticulously recorded. The study period was divided into pre-pandemic (January 2017 to 20 January 2020) and pandemic (21 January 2020 to December 2022) periods, based on the date of the first reported COVID-19 case in Taiwan.

### 2.3. Statistical Analysis

We collected and analyzed patient demographics and clinical characteristics, including the distribution of receiving hospitals, seasonal variations in EAMT, and presenting signs and symptoms. Continuous variables were presented as medians with interquartile ranges (IQRs), and categorical variables as counts and percentages. Due to the small sample size, non-parametric methods were applied: the Mann–Whitney U-test was used for continuous variables and Fisher’s exact test for categorical variables. Statistical analyses were performed using SPSS software (version 25.0; SPSS Inc., Chicago, IL, USA), and *p*-values < 0.05 were considered statistically significant.

## 3. Results

Between January 2017 and December 2022, 20 patients were transported to Taiwan before the COVID-19 pandemic and 20 patients were transported to Taiwan during the COVID-19 pandemic. The characteristics of the pediatric patients who received EAMT from the Penghu Archipelago before and during the COVID-19 pandemic are shown in Table 1. The median ages of these patients were 3 years (IQR, 0–5.00 years) before the COVID-19 pandemic and 1 year (IQR, 0–5.00 years) during the COVID-19 pandemic. Most patients were transported via rotary-wing aircraft. Two patients died at the receiving hospitals during management before the COVID-19 pandemic. Among all transported patients, the hospital length of stay at the receiving hospitals during the COVID-19 pandemic was longer than that before the COVID-19 pandemic; however, this trend was not statistically significant. The reasons for EAMT included insufficient medical equipment, lack of specialist physicians, family requests, and shortage of available hospital beds. The lack of specialist physicians showed a trend toward becoming a more prominent factor during the COVID-19 pandemic (*p* = 0.056).

During the study period, the most common medical reason for EAMT was the need for critical care medicine (35.00%), followed by neonatal disease (30.00%) and neurological disease (27.50%). The trends for medical reasons for EAMT remained similar before and during the COVID-19 pandemic. However, two patients were diagnosed with SARS-CoV-2 infections; one had encephalitis as a complication of COVID-19 and the other had croup. Changes in available medical specialties over time may have influenced the ability to provide care to pediatric patients who lived offshore (Table 2).

Compared to older children, children of preschool age (under 6 years of age) and neonates require more attention from their parents. Factors associated with EAMT of children of preschool age and neonates without trauma in Penghu County are shown in Table 3. During the study period, 27 children of preschool age and neonates without trauma were analyzed. Among these patients, 13 received EAMT before the COVID-19 pandemic and 14 received EAMT during the COVID-19 pandemic. During the COVID-19 pandemic, the length of stay at the receiving hospitals was longer than that before the COVID-19 pandemic (Figure 1). The lack of specialist physicians was the reason for EAMT for the majority of pediatric patients during the COVID-19 pandemic. The most common medical reasons for EAMT were neonatal disease (44.44%), neurological disease (37.04%), and disease requiring critical care medicine (33.33%).

No significant differences in the medical reasons for EAMT were observed before and during the COVID-19 pandemic. A comparison of patients who received EAMT during the four seasons revealed that more patients were transported during summer; however, seasonal variations were not statistically significant (Table 4). Of these patients, the majority (88.89%) were transported to Kaohsiung City. Only a few patients were transported to other cities because of distance, suggesting that distance was an important factor associated with EAMT.

## 4. Discussion

This study integrated the characteristics of pediatric patients who required EAMT from the Penghu Islands to Taiwan’s main island between January 2017 and December 2022, and focused on the periods before and during the COVID-19 pandemic. Hospital stays were longer during the COVID-19 pandemic. The increase in the hospital length of stay at the receiving hospitals during the COVID-19 pandemic may be attributed to several factors. For example, previous studies suggested that proper social distancing may help reduce pediatric hospitalizations during periods when common seasonal viral illnesses are prevalent and increased COVID-19 cases are expected [24]. Social distancing policies enacted to mitigate the COVID-19 pandemic resulted in a great decrease in the diagnosis of common infectious diseases among children [25]. The substantial decreases in hospital presentations and diagnoses highlighted how public health measures can significantly affect healthcare utilization and the prevalence of infectious diseases among pediatric populations.

The COVID-19 pandemic placed an immense burden on healthcare systems that resulted in resource reallocation and treatment delays. Additionally, stringent infection control measures may have contributed to longer hospital stays. Although a noticeable trend was observed, the limited sample size resulted in variable findings, which prevented definitive conclusions. However, this study included nearly all pediatric patients in the Penghu Islands who required EAMT.

The specific focus of this study on children of preschool age and neonates without trauma revealed that the hospital length of stay at the receiving hospitals increased significantly during the COVID-19 pandemic. This finding underscored the heightened impact of the COVID-19 pandemic on younger and vulnerable populations. The reasons for EAMT for these populations were notably influenced by the lack of specialist physicians, which became a more prominent factor during the COVID-19 pandemic. This trend suggested that resource constraints and specialist shortages have disproportionate effects on the care of children of preschool age and neonates. Although pediatric specialties, including neonatology services, have expanded recently, the addition of neonatologists during the COVID-19 pandemic did not effectively reduce EAMT. This indicated that although the availability of more specialized care is beneficial, it may not be sufficient to address all the challenges posed by a pandemic, such as resource allocation and the overall strain on healthcare systems.

The increased hospital length of stay during the COVID-19 pandemic can be attributed to several factors, such as the critical nature of care required by neonates and children of preschool age and the limited availability of specialist physicians during the COVID-19 pandemic. Furthermore, isolation protocols implemented as part of infection control measures may have contributed to prolonged hospital stays, even in patients not diagnosed with COVID-19. Delayed discharges or extended observation periods due to quarantine requirements represent an important contextual factor to consider when interpreting these findings. Additionally, the effects of the COVID-19 pandemic on healthcare resources and staff availability further exacerbated healthcare systems, thus making it difficult to provide timely and efficient care. Although no differences in the hospital stay durations of pediatric patients with common diagnoses during the COVID-19 pandemic were observed, similar studies have reported increased hospital stay durations across various regions, particularly for those whose treatment is technology-dependent and those with critical illnesses [24,26,27]. For example, previous studies have shown that critically ill pediatric patients admitted to pediatric intensive care units (PICUs) experienced prolonged lengths of stay during the pandemic [27]. Increased hospital stays were particularly noted among children with complex medical needs or those who were technology-dependent, due to delays in discharge planning, shortages of specialized personnel, and stringent infection control protocols [26]. These consistent findings of longer hospital stays reflect the global impact of the COVID-19 pandemic on healthcare systems.

For children of preschool age and neonates without trauma, longer hospital stays during the COVID-19 pandemic reached statistical significance. This observation may reflect the trends of children whose treatment is technology-dependent and those who are critically ill, suggesting that the strain of the COVID-19 pandemic on healthcare resources and services broadly affected vulnerable pediatric populations. This strain may have resulted in prolonged hospital stays, particularly for these patients. Further research is required to confirm these findings and understand their underlying mechanisms. Additionally, these findings underscore the need for robust healthcare planning and resource allocation, particularly in remote areas, such as the Penghu Islands. The prominence of factors such as insufficient medical equipment, lack of specialist physicians, family requests, and shortage of available hospital beds during the COVID-19 pandemic emphasized the vulnerabilities of healthcare systems, which need to be addressed to improve patient outcomes during future crises.

This study had several limitations. The small sample size may have limited the statistical power to detect differences between groups, and some findings may not have reached statistical significance despite observable trends. This limitation should be taken into account when interpreting the results. Due to the limited sample size and small subgroup numbers, multivariate analysis was not performed, and confidence intervals were not reported, as such estimates may lack statistical reliability and precision. Future studies with larger cohorts are warranted to enable more robust statistical analysis, including the identification of independent predictors of prolonged hospital stay through methods such as logistic regression, as well as the calculation of confidence intervals to improve the interpretability of findings. Moreover, this study relied on records from a single hospital branch. These records may not have been representative of the broader population. However, these patients comprised nearly all pediatric patients who required EAMT in the Penghu Islands during the study period. Furthermore, the categorization of medical reasons for EAMT and the potential for unrecorded variables may have affected the results. In addition to the small sample size and single-center design, this study may be subject to selection bias, as only patients transported by Tri Service General Hospital, Penghu Branch were included. Although the cohort represents nearly all pediatric EAMT cases in the Penghu Islands during the study period, the findings may not be generalizable to other remote or urban populations. Moreover, due to the retrospective nature of the study, causal inferences between the COVID-19 pandemic and the observed transport patterns or clinical outcomes cannot be definitively drawn. Larger multicenter studies should be performed to validate these findings and provide a more comprehensive understanding of the impact of the COVID-19 pandemic on pediatric patients who required EAMT. Although the distribution of medical diagnoses requiring EAMT remained relatively stable before and during the COVID-19 pandemic, systemic limitations such as reduced availability of pediatric subspecialists, strained hospital resources, and infection control protocols played a greater role in the decision to transfer patients. EAMT is essential for critically ill patients located in remote areas who require advanced management. Therefore, the exploration of strategies to enhance healthcare system resilience, such as increasing the availability of specialist physicians and medical equipment, could provide valuable insights regarding policies and practices.

## 5. Conclusions

This study highlighted the increased hospital length of stay and critical factors that affected EAMT for pediatric patients during the COVID-19 pandemic. Although the overall trend of the hospital length of stay did not reach statistical significance, a significant increase in the hospital length of stay of children of preschool age and neonates without trauma was observed. These findings emphasize the need for improved healthcare resource allocation and planning, to ensure better patient care during future public health emergencies. The consistent distributions of medical reasons for EAMT before and during the COVID-19 pandemic indicated stable underlying health issues, whereas the increased shortage of specialist physicians during the COVID-19 pandemic highlighted a crucial matter that requires improvement. Enhancing local healthcare infrastructure, particularly by decentralizing and increasing the availability of pediatric subspecialists such as neonatologists, pediatric neurologists, and critical-care providers, is essential for reducing reliance on distant referral hospitals and ensuring timely access to care. Addressing these issues will help prepare healthcare systems for future crises and ensure timely and adequate care for vulnerable pediatric populations.

## Figures and Tables

**Figure 1 healthcare-13-01450-f001:**
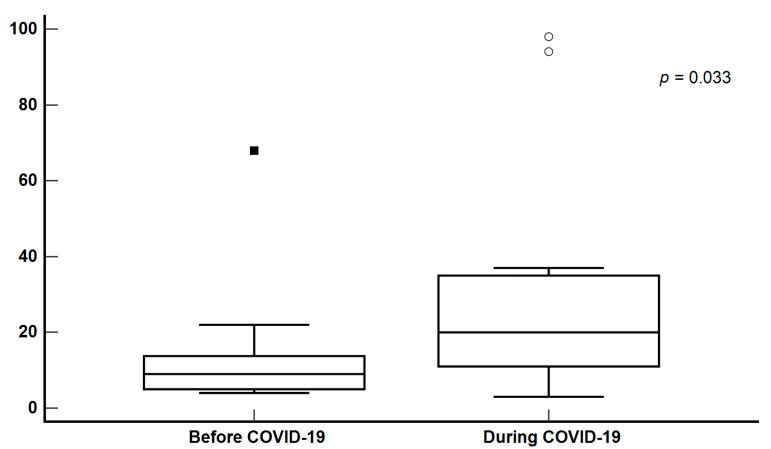
Comparison of hospital stays for non-traumatic preschool patients and neonates after air transportation. Solid squares and circles indicate outliers in different groups.

**Table 1 healthcare-13-01450-t001:** Demographic characteristics of pediatric air medical transports in the Penghu archipelago before and during the COVID-19 pandemic (patients may have more than one clinical diagnosis).

Demographics	All Cases (n = 40)	Before COVID-19 (n = 20)	During COVID-19 (n = 20)	*p*-Value
Age, median (Q1–Q3)				
<6 year	1 (0–3.00)	2 (0–3.00)	1 (0–1.50)	0.322
6 to <18 years	10 (7.50–16.50)	9 (8.25–16.25)	12.5 (8.00–16.00)	0.804
All ages	2 (0–5.00)	3 (0–5.00)	1 (0–5.00)	0.362
Gender, n (%)				
Male	28 (70.00)	15 (75.00)	13 (65.00)	0.731
Female	12 (30.00)	5 (25.00)	7 (35.00)	
Aircraft, n (%)				
rotary-wing	36 (90.00)	20 (100.00)	16 (80.00)	0.106
fixed-wing	4 (10.00)	0 (0.00)	4 (20.00)	
Hospital stay, median (Q1–Q3)	12.5 (8.00–26.50)	11 (6.00–16.50)	17 (9.00–31.00)	0.098
Reason, n (%)				
Insufficient medical equipment	26 (65.00)	15 (75.00)	11 (55.00)	0.320
Lack of specialist doctors	19 (47.50)	6 (30.00)	13 (65.00)	0.056
Family request	10 (25.00)	5 (25.00)	5 (25.00)	1.000
No beds	1 (2.50)	0 (0.00)	1 (5.00)	1.000
Disease, n (%)				
Burn injury	2 (5.00)	1 (5.00)	1 (5.00)	1.000
Traumatic injury	9 (22.50)	5 (25.00)	4 (20.00)	1.000
Neurological disease	11 (27.50)	6 (30.00)	5 (25.00)	1.000
Neonatal disease	12 (30.00)	6 (30.00)	6 (30.00)	1.000
Cardiovascular disease	6 (15.00)	4 (20.00)	2 (10.00)	0.661
Infectious disease	7 (17.50)	4 (20.00)	3 (15.00)	1.000
Genetics and Metabolism	5 (12.50)	3 (15.00)	2 (10.00)	1.000
Gastroenterology	1 (2.50)	0 (0.00)	1 (5.00)	1.000
Nephrology	3 (7.50)	1 (5.00)	2 (10.00)	1.000
Pulmonology	1 (2.50)	0 (0.00)	1 (5.00)	1.000
Hematologic disease	1 (2.50)	1 (1.00)	0 (0.00)	1.000
Critical care medicine	14 (35.00)	6 (30.00)	8 (40.00)	0.741

Note. Fisher exact test for categorical variables and Mann–Whitney U-test for continuous variables.

**Table 2 healthcare-13-01450-t002:** Pediatric specialties in Penghu across different periods.

Year	Subspecialty 1	Subspecialty 2	Subspecialty 3	Subspecialty 4
2017	Endocrinology	Cardiology	Neurology	
2018	Endocrinology	Cardiology	Neurology	
2019	Endocrinology	Cardiology	Neonatology	GI
2020	Endocrinology	Cardiology	Neonatology	GI
2021	Endocrinology	Cardiology	Neonatology	GI
2022	Endocrinology	Cardiology	Neonatology	Neurology

GI: Gastroenterology.

**Table 3 healthcare-13-01450-t003:** Factors associated with air transport of non-traumatic preschool patients and neonates in Penghu before and during the COVID-19 pandemic (patients may have more than one clinical diagnosis).

Demographics	All Cases (n = 27)	Before COVID-19 (n = 13)	During COVID-19 (n = 14)	*p*-Value
Age, median (Q1–Q3)	1 (0–2.00)	2 (0–3.00)	0.5 (0–1.00)	0.300
Gender, n (%)				
Male	17 (62.96)	9 (69.23)	8 (57.14)	0.695
Female	10 (37.04)	4 (30.77)	6 (42.86)	
Aircraft, n (%)				
rotary-wing	24 (88.89)	13 (100.00)	11 (78.57)	0.222
fixed-wing	3 (11.11)	0 (0.00)	3 (21.43)	
Hospital stay, median (Q1–Q3)	13 (8.00–26.50)	9 (5.00–13.75)	20 (11.00–35.00)	0.033
Reason, n (%)				
Insufficient medical equipment	21 (77.78)	12 (92.31)	9 (64.29)	0.165
Lack of specialist doctors	14 (51.85)	4 (30.77)	10 (71.43)	0.057
Family request	5 (18.52)	2 (15.38)	3 (21.43)	1.000
No beds	0 (0.00)	0 (0.00)	0 (0.00)	-
Disease, n (%)				
Neurological disease	10 (37.04)	6 (46.15)	4 (28.57)	0.440
Neonatal disease	12 (44.44)	6 (46.15)	6 (42.86)	1.000
Cardiovascular disease	5 (18.52)	3 (23.08)	2 (14.29)	0.648
Infectious disease	6 (22.22)	3 (23.08)	3 (21.43)	1.000
Genetics and Metabolism	5 (18.52)	3 (23.08)	2 (14.29)	0.648
Gastroenterology	1 (3.70)	0 (0.00)	1 (7.14)	1.000
Nephrology	2 (7.41)	1 (7.69)	1 (7.14)	1.000
Pulmonology	1 (3.70)	0 (0.00)	1 (7.14)	1.000
Hematologic disease	0 (0.00)	0 (0.00)	0 (0.00)	-
Critical care medicine	9 (33.33)	5 (38.46)	4 (28.57)	0.695

Note. Fisher exact test for categorical variables and Mann–Whitney U-test for continuous variables.

**Table 4 healthcare-13-01450-t004:** Seasonal variations in air transportation of patients in Penghu.

Seasons, n (%)	All Cases (n = 27)	Before COVID-19 (n = 13)	During COVID-19 (n = 14)	*p*-Value
Spring	7 (25.93)	5 (38.46)	2 (14.29)	0.209
Summer	11 (40.74)	4 (30.77)	7 (50.00)	0.440
Fall	5 (18.52)	1 (7.69)	4 (28.57)	0.326
Winter	4 (14.81)	3 (23.08)	1 (7.14)	0.326

Note. Fisher exact test for categorical variables.

## Data Availability

The original contributions presented in this study are included in the article. Further inquiries can be directed to the corresponding author.

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
