# Peer review of "Impact of the COVID-19 Pandemic on Emergency Air Medical Transport of Pediatric Patients in the Penghu Islands"

_healthcare, 2025, doi:10.3390/healthcare13121450_

Round 1
Reviewer 1 Report
Comments and Suggestions for Authors
Thank you for the work you have done on this study. You were certainly in an interesting position for helath care.Can you explain this statement please - did COVID, lack of beds and equipment due to COVID not impact on the need to transfer to outside health care centers. No significant differences in the medical reasons for EAMT were observed before and during the COVID-19 pandemic. I think this will help with your paper.
Reviewer 2 Report
Comments and Suggestions for Authors
I am grateful for the opportunity to review this manuscript, which addresses a sensitive and highly relevant topic: the impact of the COVID-19 pandemic on pediatric emergency air medical transport in a remote island region.
This is the first study to specifically assess the impact of the pandemic on pediatric EAMT in the Penghu Islands, a geographic region with unique structural and logistical challenges, and the analysis contributes to the understanding of disparities in access to specialized care in geographically isolated contexts.
Below I present some points that merit review:
- Page 2, lines 26-30: Specify what type of medical documentation was reviewed (e.g., electronic medical records, transport sheets, air ambulance records).
- Page 3, lines 7-10: No justification for sample size (n = 40).
- Include justification for sample size or recognized limitation of small number of cases.
- I suggest mentioning the risk of bias by controlling for clinical severity, response time, or standardized criteria to indicate transport.
- The criteria for defining the "pre-pandemic" and "during the pandemic" periods are not clearly described. I suggest that the cut-off date be explicitly stated (recommended: March 2020, the official date of the WHO pandemic declaration).
Statistics: - Page 3, lines 20-30: The statistical description is superficial.
- Provide 95% confidence intervals for estimates.
- No multivariate analysis was performed; I recommend considering logistic regression to explore predictors of longer hospital stays.
- Comparisons between periods lack statistical power due to small sample size, but this is not mentioned as a limitation.
RESULTS: The data on the increase in the average number of days spent in hospital during the pandemic among neonates and preschool children were statistically significant (p = 0.033, Table 3) - a relevant and well highlighted point.
- Page 4, Table 1:
- Present medians and IQRs directly in tables (use "median (Q1-Q3)" as headline).
- The p-value = 0.056 for "lack of specialist" should be reported as a statistical trend.
- Page 5, Table 3:
- It is unclear whether the diagnoses are mutually exclusive (could a patient have more than one clinical reason?). Specify this in the title.
DISCUSSION: - Page 6-7:
- Methodological limitations are not discussed critically enough. I suggest including a discussion of selection bias, limitations of generalizability, and the impossibility of causal inference.
- The possibility of confounding by increased length of hospital stay due to isolation protocols was not discussed, which is relevant.
CONCLUSION: In the conclusion. I propose to emphasize the need for strategies to decentralize specialists to remote islands.
Reviewer 3 Report
Comments and Suggestions for Authors
Dear Author,
Thank you for your efforts in addressing such a vital topic during the COVID-19 pandemic.
Please consider the following comments for revision:
Abstract
- Kindly include the measurement tools used in the methods section.
- The results and conclusion need to be rewritten to provide more specific and significant information.
Introduction
- Add the significance of the study and elaborate on the importance of medical emergency preparedness for future pandemics.
- Please clarify the rationale for focusing primarily on pediatric patients.
Materials and Methods
- Ensure that the study design is clearly reflected in the abstract.
Discussion
- Please compare the results with other regional and international studies.
- Include practical implications for future pandemics, particularly in terms of improving emergency responses and saving patient lives.
Best regards.
